# Dual Learning for Machine Translation

**Di He**[1,*]**, Yingce Xia**[2,*]**, Tao Qin**[3]**, Liwei Wang**[1]**, Nenghai Yu**[2]**, Tie-Yan Liu**[3]**, Wei-Ying Ma**[3]

[1]Key Laboratory of Machine Perception (MOE), School of EECS, Peking University
[2]University of Science and Technology of China    [3]Microsoft Research
[1]{dih,wanglw}@cis.pku.edu.cn; [2]xiayingc@mail.ustc.edu.cn; [2]ynh@ustc.edu.cn
[3]{taoqin,tie-yan.liu,wyma}@microsoft.com

## Abstract

While neural machine translation (NMT) is making good progress in the past two years, tens of millions of bilingual sentence pairs are needed for its training. However, human labeling is very costly. To tackle this training data bottleneck, we develop a dual-learning mechanism, which can enable an NMT system to automatically learn from unlabeled data through a dual-learning game. This mechanism is inspired by the following observation: any machine translation task has a dual task, e.g., English-to-French translation (primal) versus French-to-English translation (dual); the primal and dual tasks can form a closed loop, and generate informative feedback signals to train the translation models, even if without the involvement of a human labeler. In the dual-learning mechanism, we use one agent to represent the model for the primal task and the other agent to represent the model for the dual task, then ask them to teach each other through a reinforcement learning process. Based on the feedback signals generated during this process (e.g., the language-model likelihood of the output of a model, and the reconstruction error of the original sentence after the primal and dual translations), we can iteratively update the two models until convergence (e.g., using the policy gradient methods). We call the corresponding approach to neural machine translation *dual-NMT*. Experiments show that dual-NMT works very well on English↔French translation; especially, by learning from monolingual data (with 10% bilingual data for warm start), it achieves a comparable accuracy to NMT trained from the full bilingual data for the French-to-English translation task.

## 1   Introduction

State-of-the-art machine translation (MT) systems, including both the phrase-based statistical translation approaches [6, 3, 12] and the recently emerged neural networks based translation approaches [1, 5], heavily rely on aligned parallel training corpora. However, such parallel data are costly to collect in practice and thus are usually limited in scale, which may constrain the related research and applications.

Given that there exist almost unlimited monolingual data in the Web, it is very natural to leverage them to boost the performance of MT systems. Actually different methods have been proposed for this purpose, which can be roughly classified into two categories. In the first category [2, 4], monolingual corpora in the target language are used to train a language model, which is then integrated with the MT models trained from parallel bilingual corpora to improve the translation quality. In the second category [14, 11], pseudo bilingual sentence pairs are generated from monolingual data by using the

translation model trained from aligned parallel corpora, and then these pseudo bilingual sentence pairs are used to enlarge the training data for subsequent learning. While the above methods could improve the MT performance to some extent, they still suffer from certain limitations. The methods in the first category only use the monolingual data to train language models, but do not fundamentally address the shortage of parallel training data. Although the methods in the second category can enlarge the parallel training data, there is no guarantee/control on the quality of the pseudo bilingual sentence pairs.

In this paper, we propose a dual-learning mechanism that can leverage monolingual data (in both the source and target languages) in a more effective way. By using our proposed mechanism, these monolingual data can play a similar role to the parallel bilingual data, and significantly reduce the requirement on parallel bilingual data during the training process. Specifically, the dual-learning mechanism for MT can be described as the following two-agent communication game.

1. The first agent, who only understands language A, sends a message in language A to the second agent through a noisy channel, which converts the message from language A to language B using a translation model.

2. The second agent, who only understands language B, receives the translated message in language B. She checks the message and notifies the first agent whether it is a natural sentence in language B (note that the second agent may not be able to verify the correctness of the translation since the original message is invisible to her). Then she sends the received message back to the first agent through another noisy channel, which converts the received message from language B back to language A using another translation model.

3. After receiving the message from the second agent, the first agent checks it and notifies the second agent whether the message she receives is consistent with her original message. Through the feedback, both agents will know whether the two communication channels (and thus the two translation models) perform well and can improve them accordingly.

4. The game can also be started from the second agent with an original message in language B, and then the two agents will go through a symmetric process and improve the two channels (translation models) according to the feedback.

It is easy to see from the above descriptions, although the two agents may not have aligned bilingual corpora, they can still get feedback about the quality of the two translation models and collectively improve the models based on the feedback. This game can be played for an arbitrary number of rounds, and the two translation models will get improved through this reinforcement procedure (e.g., by means of the policy gradient methods). In this way, we develop a general learning framework for training machine translation models through a dual-learning game.

The dual learning mechanism has several distinguishing features. First, we train translation models from unlabeled data through reinforcement learning. Our work significantly reduces the requirement on the aligned bilingual data, and it opens a new window to learn to translate from scratch (i.e., even without using any parallel data). Experimental results show that our method is very promising.

Second, we demonstrate the power of deep reinforcement learning (DRL) for complex real-world applications, rather than just games. Deep reinforcement learning has drawn great attention in recent years. However, most of them today focus on video or board games, and it remains a challenge to enable DRL for more complicated applications whose rules are not pre-defined and where there is no explicit reward signals. Dual learning provides a promising way to extract reward signals for reinforcement learning in real-world applications like machine translation.

The remaining parts of the paper are organized as follows. In Section 2, we briefly review the literature of neural machine translation. After that, we introduce our dual-learning algorithm for neural machine translation. The experimental results are provided and discussed in Section 4. We extend the breadth and depth of dual learning in Section 5 and discuss future work in the last section.

## 2   Background: Neural Machine Translation

In principle, our dual-learning framework can be applied to both phrase-based statistical machine translation and neural machine translation. In this paper, we focus on the latter one, i.e., neural

machine translation (NMT), due to its simplicity as an end-to-end system, without suffering from human crafted engineering [5].

Neural machine translation systems are typically implemented with a Recurrent Neural Network (RN-N) based encoder-decoder framework. Such a framework learns a probabilistic mapping $P(y|x)$ from a source language sentence $x = \{x_1, x_2, ..., x_{T_x}\}$ to a target language sentence $y = \{y_1, y_2, ..., y_{T_y}\}$, in which $x_i$ and $y_t$ are the $i$-th and $t$-th words for sentences $x$ and $y$ respectively.

To be more concrete, the encoder of NMT reads the source sentence $x$ and generates $T_x$ hidden states by an RNN:

$$h_i = f(h_{i-1}, x_i) \tag{1}$$

in which $h_i$ is the hidden state at time $i$, and function $f$ is the recurrent unit such as Long Short-Term Memory (LSTM) unit [12] or Gated Recurrent Unit (GRU) [3]. Afterwards, the decoder of NMT computes the conditional probability of each target word $y_t$ given its proceeding words $y_{<t}$, as well as the source sentence, i.e., $P(y_t|y_{<t}, x)$, which is then used to specify $P(y|x)$ according to the probability chain rule. $P(y_t|y_{<t}, x)$ is given as:

$$P(y_t|y_{<t}, x) \propto \exp(y_t; r_t, c_t) \tag{2}$$
$$r_t = g(r_{t-1}, y_{t-1}, c_t) \tag{3}$$
$$c_t = q(r_{t-1}, h_1, \cdots, h_{T_x}) \tag{4}$$

where $r_t$ is the decoder RNN hidden state at time $t$, similarly computed by an LSTM or GRU, and $c_t$ denotes the contextual information in generating word $y_t$ according to different encoder hidden states. $c_t$ can be a 'global' signal summarizing sentence $x$ [3, 12], e.g., $c_1 = \cdots = c_{T_y} = h_{T_x}$, or 'local' signal implemented by an attention mechanism [1], e.g., $c_t = \sum_{i=1}^{T_x} \alpha_i h_i$, $\alpha_i = \frac{\exp\{a(h_i, r_{t-1})\}}{\sum_j \exp\{a(h_j, r_{t-1})\}}$, where $a(\cdot, \cdot)$ is a feed-forward neural network.

We denote all the parameters to be optimized in the neural network as $\Theta$ and denote $D$ as the dataset that contains source-target sentence pairs for training. Then the learning objective is to seek the optimal parameters $\Theta^*$:

$$\Theta^* = \underset{\Theta}{\arg\max} \sum_{(x,y) \in D} \sum_{t=1}^{T_y} \log P(y_t|y_{<t}, x; \Theta) \tag{5}$$

## 3 Dual Learning for Neural Machine Translation

In this section, we present the dual-learning mechanism for neural machine translation. Noticing that MT can (always) happen in dual directions, we first design a two-agent game with a forward translation step and a backward translation step, which can provide quality feedback to the dual translation models even using monolingual data only. Then we propose a dual-learning algorithm, called *dual-NMT*, to improve the two translation models based on the quality feedback provided in the game.

Consider two monolingual corpora $D_A$ and $D_B$ which contain sentences from language $A$ and $B$ respectively. Please note these two corpora are not necessarily aligned with each other, and they may even have no topical relationship with each other at all. Suppose we have two (weak) translation models that can translate sentences from $A$ to $B$ and verse visa. Our goal is to improve the accuracy of the two models by using monolingual corpora instead of parallel corpora. Our basic idea is to leverage the duality of the two translation models. Starting from a sentence in any monolingual data, we first translate it forward to the other language and then further translate backward to the original language. By evaluating this two-hop translation results, we will get a sense about the quality of the two translation models, and be able to improve them accordingly. This process can be iterated for many rounds until both translation models converge.

Suppose corpus $D_A$ contains $N_A$ sentences, and $D_B$ contains $N_B$ sentences. Denote $P(.|s; \Theta_{AB})$ and $P(.|s; \Theta_{BA})$ as two neural translation models, where $\Theta_{AB}$ and $\Theta_{BA}$ are their parameters (as described in Section 2).

Assume we already have two well-trained language models $LM_A(.)$ and $LM_B(.)$ (which are easy to obtain since they only require monolingual data), each of which takes a sentence as input and outputs

---

**Algorithm 1** The dual-learning algorithm

---

1: **Input**: Monolingual corpora $D_A$ and $D_B$, initial translation models $\Theta_{AB}$ and $\Theta_{BA}$, language models $LM_A$ and $LM_B$, $\alpha$, beam search size $K$, learning rates $\gamma_{1,t}, \gamma_{2,t}$ .
2: **repeat**
3:     $t = t + 1$.
4:     Sample sentence $s_A$ and $s_B$ from $D_A$ and $D_B$ respectively.
5:     Set $s = s_A$.                             ▷ *Model update for the game beginning from A.*
6:     Generate $K$ sentences $s_{mid,1}, \ldots, s_{mid,K}$ using beam search according to translation model $P(.|s; \Theta_{AB})$.
7:     **for** $k = 1, \ldots, K$ **do**
8:         Set the language-model reward for the $k$th sampled sentence as $r_{1,k} = LM_B(s_{mid,k})$.
9:         Set the communication reward for the $k$th sampled sentence as $r_{2,k} = \log P(s|s_{mid,k}; \Theta_{BA})$.
10:        Set the total reward of the $k$th sample as $r_k = \alpha r_{1,k} + (1 - \alpha) r_{2,k}$.
11:    **end for**
12:    Compute the stochastic gradient of $\Theta_{AB}$:

$$\nabla_{\Theta_{AB}} \hat{E}[r] = \frac{1}{K} \sum_{k=1}^{K} [r_k \nabla_{\Theta_{AB}} \log P(s_{mid,k}|s; \Theta_{AB})].$$

13:    Compute the stochastic gradient of $\Theta_{BA}$:

$$\nabla_{\Theta_{BA}} \hat{E}[r] = \frac{1}{K} \sum_{k=1}^{K} [(1 - \alpha) \nabla_{\Theta_{BA}} \log P(s|s_{mid,k}; \Theta_{BA})].$$

14:    Model updates:

$$\Theta_{AB} \leftarrow \Theta_{AB} + \gamma_{1,t} \nabla_{\Theta_{AB}} \hat{E}[r], \Theta_{BA} \leftarrow \Theta_{BA} + \gamma_{2,t} \nabla_{\Theta_{BA}} \hat{E}[r].$$

15:    Set $s = s_B$.                             ▷ *Model update for the game beginning from B.*
16:    Go through line 6 to line 14 symmetrically.
17: **until** convergence

---

a real value to indicate how confident the sentence is a natural sentence in its own language. Here the language models can be trained either using other resources, or just using the monolingual data $D_A$ and $D_B$.

For a game beginning with sentence $s$ in $D_A$, denote $s_{mid}$ as the middle translation output. This middle step has an immediate reward $r_1 = LM_B(s_{mid})$, indicating how natural the output sentence is in language B. Given the middle translation output $s_{mid}$, we use the log probability of $s$ recovered from $s_{mid}$ as the reward of the communication (we will use *reconstruction* and *communication* interchangeably). Mathematically, reward $r_2 = \log P(s|s_{mid}; \Theta_{BA})$.

We simply adopt a linear combination of the LM reward and communication reward as the total reward, e.g., $r = \alpha r_1 + (1 - \alpha) r_2$, where $\alpha$ is a hyper-parameter. As the reward of the game can be considered as a function of $s$, $s_{mid}$ and translation models $\Theta_{AB}$ and $\Theta_{BA}$, we can optimize the parameters in the translation models through *policy gradient methods* for reward maximization, as widely used in reinforcement learning [13].

We sample $s_{mid}$ according to the translation model $P(.|s; \Theta_{AB})$. Then we compute the gradient of the expected reward $E[r]$ with respect to parameters $\Theta_{AB}$ and $\Theta_{BA}$. According to the policy gradient theorem [13], it is easy to verify that

$$\nabla_{\Theta_{BA}} E[r] = E[(1 - \alpha) \nabla_{\Theta_{BA}} \log P(s|s_{mid}; \Theta_{BA})] \tag{6}$$

$$\nabla_{\Theta_{AB}} E[r] = E[r \nabla_{\Theta_{AB}} \log P(s_{mid}|s; \Theta_{AB})] \tag{7}$$

in which the expectation is taken over $s_{mid}$.

Based on Eqn.(6) and (7), we can adopt any sampling approach to estimate the expected gradient. Considering that random sampling brings very large variance and sometimes unreasonable results in

Table 1: Translation results of En↔Fr task. The results of the experiments using all the parallel data for training are provided in the first two columns (marked by "Large"), and the results using 10% parallel data for training are in the last two columns (marked by "Small").

| | En→Fr (Large) | Fr→En (Large) | En→Fr (Small) | Fr→En (Small) |
|---|---|---|---|---|
| NMT | 29.92 | 27.49 | 25.32 | 22.27 |
| pseudo-NMT | 30.40 | 27.66 | 25.63 | 23.24 |
| dual-NMT | **32.06** | **29.78** | **28.73** | **27.50** |

machine translation [9, 12, 10], we use beam search [12] to obtain more meaningful results (more reasonable middle translation outputs) for gradient computation, i.e., we greedily generate top-$K$ high-probability middle translation outputs, and use the averaged value on the beam search results to approximate the true gradient. If the game begins with sentence $s$ in $D_B$, the computation of the gradient is just symmetric and we omit it here.

The game can be repeated for many rounds. In each round, one sentence is sampled from $D_A$ and one from $D_B$, and we update the two models according to the game beginning with the two sentences respectively. The details of this process are given in Algorithm 1.

# 4 Experiments

We conducted a set of experiments to test the proposed dual-learning mechanism for neural machine translation.

## 4.1 Settings

We compared our dual-NMT approach with two baselines: the standard neural machine translation [1] (NMT for short), and a recent NMT-based method [11] which generates pseudo bilingual sentence pairs from monolingual corpora to assist training (pseudo-NMT for short). We leverage a tutorial NMT system implemented by Theano for all the experiments. [2]

We evaluated our algorithm on the translation task of a pair of languages: English→French (En→Fr) and French→English (Fr→En). In detail, we used the same bilingual corpora from WMT'14 as used in [1, 5], which contains 12M sentence pairs extracting from five datasets: Europarl v7, Common Crawl corpus, UN corpus, News Commentary, and $10^9$French-English corpus. Following common practices, we concatenated newstest2012 and newstest2013 as the validation set, and used newstest2014 as the testing set. We used the "News Crawl: articles from 2012" provided by WMT'14 as monolingual data.

We used the GRU networks and followed the practice in [1] to set experimental parameters. For each language, we constructed the vocabulary with the most common 30K words in the parallel corpora, and out-of-vocabulary words were replaced with a special token <UNK>. For monolingual corpora, we removed the sentences containing at least one out-of-vocabulary words. Each word was projected into a continuous vector space of 620 dimensions, and the dimension of the recurrent unit was 1000. We removed sentences with more than 50 words from the training set. Batch size was set as 80 with 20 batches pre-fetched and sorted by sentence lengths.

For the baseline NMT model, we exactly followed the settings reported in [1]. For the baseline pseudo-NMT [11], we used the trained NMT model to generate pseudo bilingual sentence pairs from monolingual data, removed the sentences with more than 50 words, merged the generated data with the original parallel training data, and then trained the model for testing. Each of the baseline models was trained with AdaDelta [15] on K40m GPU until their performances stopped to improve on the validation set.

Our method needs a language model for each language. We trained an RNN based language model [7] for each language using its corresponding monolingual corpus. Then the language model was

Table 2: Reconstruction performance of En↔Fr task

|  | En→Fr→En (L) | Fr→En→Fr (L) | En→Fr→En (S) | Fr→En→Fr (S) |
|---|---|---|---|---|
| NMT | 39.92 | 45.05 | 28.28 | 32.63 |
| pseudo-NMT | 38.15 | 45.41 | 30.07 | 34.54 |
| dual-NMT | **51.84** | **54.65** | **48.94** | **50.38** |

fixed and the log likelihood of a received message was used to reward the communication channel (i.e., the translation model) in our experiments.

While playing the game, we initialized the channels using warm-start translation models (e.g., trained from bilingual data corpora), and see whether dual-NMT can effectively improve the machine translation accuracy. In our experiments, in order to smoothly transit from the initial model trained from bilingual data to the model training purely from monolingual data, we adopted the following soft-landing strategy. At the very beginning of the dual learning process, for each mini batch, we used half sentences from monolingual data and half sentences from bilingual data (sampled from the dataset used to train the initial model). The objective was to maximize the weighted sum of the reward based on monolingual data defined in Section 3 and the likelihood on bilingual data defined in Section 2. When the training process went on, we gradually increased the percentage of monolingual sentences in the mini batch, until no bilingual data were used at all. Specifically, we tested two settings in our experiments:

- In the first setting (referred to *Large*), we used all the 12M bilingual sentences pairs during the soft-landing process. That is, the warm start model was learnt based on full bilingual data.

- In the second setting (referred to *Small*), we randomly sampled 10% of the 12M bilingual sentences pairs and used them during the soft-landing process.

For each of the settings we trained our dual-NMT algorithm for one week. We set the beam search size to be 2 in the middle translation process. All the hyperparameters in the experiments were set by cross validation.We used the BLEU score [8] as the evaluation metric, which are computed by the *multi-bleu.perl* script[3]. Following the common practice, during testing we used beam search [12] with beam size of 12 for all the algorithms as in many previous works.

## 4.2 Results and Analysis

We report the experimental results in this section. Recall that the two baselines for English→French and French→English are trained separately while our dual-NMT conducts joint training. We summarize the overall performances in Table 1 and plot the BLEU scores with respect to the length of source sentences in Figure 1.

From Table 1 we can see that our dual-NMT algorithm outperforms the baseline algorithms in all the settings. For the translation from English to French, dual-NMT outperforms the baseline NMT by about 2.1/3.4 points for the first/second warm start setting, and outperforms pseudo-NMT by about 1.7/3.1 points for both settings. For the translation from French to English, the improvement is more significant: our dual-NMT outperforms NMT by about 2.3/5.2 points for the first/second warm start setting, and outperforms pseudo-NMT by about 2.1/4.3 points for both settings. Surprisingly, with only 10% bilingual data, dual-NMT achieves comparable translation accuracy as vanilla NMT using 100% bilingual data for the Fr→En task. These results demonstrate the effectiveness of our dual-NMT algorithm. Furthermore, we have the following observations:

- Although pseudo-NMT outperforms NMT, its improvements are not very significant. Our hypothesis is that the quality of pseudo bilingual sentence pairs generated from the monolingual data is not very good, which limits the performance gain of pseudo-NMT. One might need to carefully select and filter the generated pseudo bilingual sentence pairs to get better performance for pseudo-NMT.

Table 3: Cases study of the translation-back-translation (TBT) performance during dual-NMT training

| | Translation-back-translation results before dual-NMT training | Translation-back-translation results after dual-NMT training |
|---|---|---|
| Source (En) | The majority of the growth in the years to come will come from its liquefied natural gas schemes in Australia. | |
| En→Fr | La plus grande partie de la crois--sance des années à venir viendra de ses systèmes de gaz naturel liquéfié en Australie . | La majorité de la croissance dans les années à venir viendra de ses régimes de gaz naturel liquéfié en Australie . |
| En→Fr→En | Most of the growth of future years will come from its liquefied natural gas systems in Australia . | The majority of growth in the coming years will come from its liquefied natural gas systems in Australia . |
| Source (Fr) | Il précise que &quot; les deux cas identifiés en mai 2013 restent donc les deux seuls cas confirmés en France à ce jour " . | |
| Fr→En | He noted that " the two cases identified in May 2013 therefore remain the only two two confirmed cases in France to date " . | He states that " the two cases identified in May 2013 remain the only two confirmed cases in France to date " |
| Fr→En→Fr | Il a noté que " les deux cas identifiésen mai 2013 demeurent donc les deux seuls deux deux cas confirmés en France à ce jour " | Il précise que " les deux cas identifiés en mai 2013 restent les seuls deux cas confirmés en France à ce jour ". |

- When the parallel bilingual data are small, dual-NMT makes larger improvement. This shows that the dual-learning mechanism makes very good utilization of monolingual data. Thus we expect dual-NMT will be more helpful for language pairs with smaller labeled parallel data. Dual-NMT opens a new window to learn to translate from scratch.

We plot BLEU scores with respect to the length of source sentences in Figure 1. From the figure, we can see that our dual-NMT algorithm outperforms the baseline algorithms in all the ranges of length.

We make some deep studies on our dual-NMT algorithm in Table 2. We study the self-reconstruction performance of the algorithms: For each sentence in the test set, we translated it forth and back using the models and then checked how close the back translated sentence is to the original sentence using the BLEU score. We also used beam search to generate all the translation results. It can be easily seen from Table 2 that the self-reconstruction BLEU scores of our dual-NMT are much higher than NMT and pseudo-NMT. In particular, our proposed method outperforms NMT by about 11.9/9.6 points when using warm-start model trained on large parallel data, and outperforms NMT for about 20.7/17.8 points when using the warm-start model trained on 10% parallel data.

We list several example sentences in Table 3 to compare the self-reconstruction results of models before and after dual learning. It is quite clear that after dual learning, the reconstruction is largely improved for both directions, i.e., English→French→English and French→English→French.

To summarize, all the results show that the dual-learning mechanism is promising and better utilizes the monolingual data.

## 5 Extensions

In this section, we discuss the possible extensions of our proposed dual learning mechanism.

First, although we have focused on machine translation in this work, the basic idea of dual learning is generally applicable: as long as two tasks are in dual form, we can apply the dual-learning mechanism to simultaneously learn both tasks from unlabeled data using reinforcement learning algorithms. Actually, many AI tasks are naturally in dual form, for example, speech recognition versus text to speech, image caption versus image generation, question answering versus question generation (e.g., Jeopardy!), search (matching queries to documents) versus keyword extraction (extracting keywords/queries for documents), so on and so forth. It would very be interesting to design and test dual-learning algorithms for more dual tasks beyond machine translation.

Second, although we have focused on dual learning on two tasks, our technology is not restricted to two tasks only. Actually, our key idea is to form a closed loop so that we can extract feedback signals by comparing the original input data with the final output data. Therefore, if more than two associated tasks can form a closed loop, we can apply our technology to improve the model in each task from unlabeled data. For example, for an English sentence $x$, we can first translate it to a Chinese sentence $y$, then translate $y$ to a French sentence $z$, and finally translate $z$ back to an English sentence $x'$. The similarity between $x$ and $x'$ can indicate the effectiveness of the three translation models in the loop, and we can once again apply the policy gradient methods to update and improve these models based on the feedback signals during the loop. We would like to name this generalized dual learning as *close-loop learning*, and will test its effectiveness in the future.

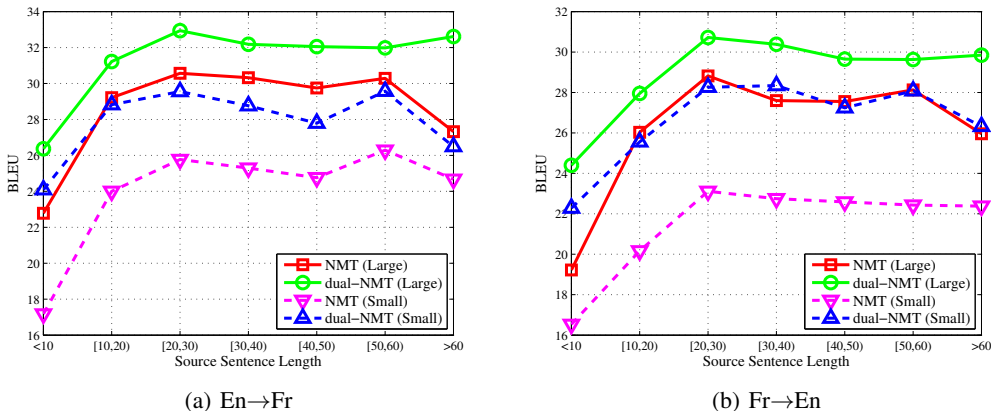

(a) En→Fr          (b) Fr→En

Figure 1: BLEU scores w.r.t lengths of source sentences

# 6   Future Work

We plan to explore the following directions in the future. First, in the experiments we used bilingual data to warm start the training of dual-NMT. A more exciting direction is to learn from scratch, i.e., to learn translations directly from monolingual data of two languages (maybe plus lexical dictionary). Second, our dual-NMT was based on NMT systems in this work. Our basic idea can also be applied to phrase-based SMT systems and we will look into this direction. Third, we only considered a pair of languages in this paper. We will extend our approach to jointly train multiple translation models for a tuple of $3+$ languages using monolingual data.

# Acknowledgement

This work was partially supported by National Basic Research Program of China (973 Program) (grant no. 2015CB352502), NSFC (61573026) and the MOE–Microsoft Key Laboratory of Statistics and Machine Learning, Peking University. We would like to thank Yiren Wang, Fei Tian, Li Zhao and Wei Chen for helpful discussions, and the anonymous reviewers for their valuable comments on our paper.

## Footnotes

*The first two authors contributed equally to this work. This work was conducted when the second author was visiting Microsoft Research Asia.

[2]*dl4mt-tutorial*: https://github.com/nyu-dl

[3]https://github.com/moses-smt/mosesdecoder/blob/master/scripts/generic/multi-bleu.perl

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
