[Reviews · NeurIPS 2016]

Reviewer 1

Summary

This paper proposes a method to benefit from monolingual corpora to improve a NMT system. The framework relies on the idea of reinforcement learning to provide feedback to 2 NMT systems (one in each direction for a given language pair). The method is well described and yields impressive gains. The paper is overall well written. While there's nothing really new in terms of machine learning, this work merges different novelties to create an efficient framework that will create new perspectives in NLP.

Qualitative Assessment

A general remark. Could feedback function also use the language model corresponding to the source sentence. I don't know how to that, but this could acts as an indicator of the sentence difficulty and therefore mitigate the feedback of wired source sentence. Could you also motivate why you discards sentence with at least one unknown word. This could be handled in your framework. Did you try something else ? L254. Avoid the repetition of in table 3. L255. "For" must be in lowercase Page 9: Highlight For arxiv papers, you could cite the published versions.

Confidence in this Review

3-Expert (read the paper in detail, know the area, quite certain of my opinion)


Reviewer 2

Summary

This paper presents a model called "communication-based machine translation (CMT), where translation models for the two translation directions are improved jointly by forth- and back-translating monolingual data and collecting feedback after each sampled translation. The goal is to leverage monolingual data in an effective way for improving machine translation based on bilingual corpora. The model is evaluated on two language pairs and compared to a standard neural MT system and a system that is additionally trained on pseudo-bilingual data.

Qualitative Assessment

The same goal has been pursued by e.g. including a language model on the target side for re-scoring translations [1] or include it into the MT system via deep or shallow fusion [2]. The paper does not sufficiently review the work that has been done in this direction and only focuses on the recent work by Sennrich et al. Since the goal of exploiting monolingual data for MT has been in the focus of many works, more empirical comparisons are needed to demonstrate the superiority of their system. It would have been easy to e.g. use the same data set as in [2] to make a direct comparison. Also, there has been work on the unsupervised training of noisy-channel models [3] which needs to be mentioned. In addition, there are no comparisons to other communication-based learning scenarios in reinforcement learning. Instead of repeatingly describing the basic idea of having two players communicating via two translation models, the paper would benefit from a more careful and thorough review and comparisons of previous work. The authors conclude that this model gives new directions for learning the translation model from scratch from monolingual data, but I personally doubt that this will work, since both Ranzato et. al. and Shen et al. [4] who learn translation models from reinforcement learning observed that a warm-start model is required, because the feedback is too weak. In line 116 they assume that translation starts with "weak" models, but in the end they use "well-trained" (l 152) translation models. The only experiment which suppports that the improvement is really large with monolingual data and a "weak" model is Fr->En. Also the extension to multiple languages in a translation chain is not as straight-forward as described, because the source of translation errors cannot be detected that easily anymore, so the updates might get very noisy. Although the metaphor of the two-players-game is nice, it does not align well with the actual algorithm. The assumption is that each player only understands one language, so scoring via a language model is possible, but the communication reward requires knowledge of the actual translation model, which is unavailable for the player in the described game. In order to accept the experiments as supporting their model, more details about the models (number of training instances - monolingual and bilingual) need to be reported and better comparisons have to be made (on the original data from the referred work and state-of-the-art as upper bound). To put it in a nutshell, they present a nice (but not so novel) idea, but the paper lacks sufficient (experimental) comparisons to closely related work for me to accept it. [1] Holger Schwenk. 2012. Continuous space translation models for phrase-based statistical machine translation. In Martin Kay and Christian Boitet, editors, Proceedings of the 24th International Conference on Computational Linguistics (COLIN), pages 1071–1080. Indian Institute of Technology Bombay. [2] Gulcehre, C., Firat, O., Xu, K., Cho, K., Barrault, L., Lin, H. C., ... & Bengio, Y. (2015). On using monolingual corpora in neural machine translation. arXiv preprint arXiv:1503.03535. [3] Mylonakis, Markos, Khalil Sima'an, and Rebecca Hwa. "Unsupervised estimation for noisy-channel models." Proceedings of the 24th international conference on Machine learning. ACM, 2007. [4] Shen, S., Cheng, Y., He, Z., He, W., Wu, H., Sun, M., & Liu, Y. (2015). Minimum risk training for neural machine translation. arXiv preprint arXiv:1512.02433.

Confidence in this Review

2-Confident (read it all; understood it all reasonably well)


Reviewer 3

Summary

The paper describes a way of jointly training a pair of machine translation systems using monolingual data. This is described as a kind of communication game, but it seems like a variation on the idea of autoencoders (broadly construed). Essentially, we have models of p(X|Y) and p(Y|X) and we train them jointly on independent samples from X (or Y) and we learn to predict a sample X from itself using p(X|Y)p(Y|X). (Hence Y can be seen as a "representation" of X; again, this is construing the autoencoder idea quite broadly). The technical ideas here are relatively simple but it is a neat idea that may actually be quite useful. The experiments suggest that this is a promising direction, although they seem preliminary; there are many natural questions left unanswered.

Qualitative Assessment

The technical ideas are simple enough to understand, to the point where I felt their explanation was actually rather belabored: the basic idea is repeated in the abstract, on p. 2 (line 47) and p. 3 (line 124). Rather than repeatedly explain this simple idea, I think the paper would have been stronger if it included a more thorough empirical explanation, i.e. showing learning curves, the effects of initializing with various amounts of data, and an analysis of the results: in what respects do systems trained this way improve? It obviously can't be in vocabulary (since the vocabulary of these systems is capped at 30K words). Is it word choice, word order, or something else?

Confidence in this Review

3-Expert (read the paper in detail, know the area, quite certain of my opinion)


Reviewer 4

Summary

This paper formulates machine translation as a bidirectional reinforcement learning problem using RNN translation models and two corpora (which do not have to be aligned). The objective is to minimize the linear combination of two losses from 1) directly the language corpora and 2) how consistent the recovered message is (communication).

Qualitative Assessment

This is a new idea that allows for the possibility of translation without aligned parallel corpora. I think that this could also open up new ideas in other fields as well. Therefore, the novelty factor is there. I would like to see some more data on the selection of $\alpha$, which is the weighting factor of the two losses. For example, why was alpha = 0.01? Was it because the strength of the communication loss was naturally much higher? Or was communication consistency more of an impact?

Confidence in this Review

2-Confident (read it all; understood it all reasonably well)


Reviewer 5

Summary

This paper proposes a reinforcement learning based method for machine translation by leveraging monolingual data and two-player communication. There are two innovations in the paper, firstly, the proposed model jointly train two dual translation models in one framework. Secondly, the translation models take unpaired data as inputs and learn the translation model through reinforcement.

Qualitative Assessment

The proposed framework is novel and the idea is explained in details. The paper is well-written and I really enjoy reading the paper. Though, there is a question regarding the algorithm. Can you explain why there are $\frac{1}{K}$ in both line 12 and line 13 in Algorithm 1?

Confidence in this Review

2-Confident (read it all; understood it all reasonably well)


Reviewer 6

Summary

It's a well written paper with interesting ideas: considering machine translation as a communication game between two parties who only know their own respective language.

Qualitative Assessment

The following points were raised: 1. The method doesn't seem fully developed from a theoretical perspective. Would be great to see more theoretical analysis or discussion. 2. There are many missing details. For example, what is the exact one-way translation model used and how is its architecture/parameters determined? What is the exact model for evaluating the quality of pseudo-translations? Details like those are important. 3. In Algorithm 1 there is a scheduling that is alternating between A to B and B to A, one sentence by another. How effective is this scheduling and can it be made better? E.g. when sampling sentence from B one can use one that show some overlap with the pseudo-translation of the sentence from A in the last round. 4. The experiment is somewhat disappointing. Because it is monolingual translation, the proposed method could be applied to much larger datasets but the authors only tested on conventional small ones. How about train on the entire Wikipedia of English and French? 5. Another interesting question is how the method compare to training on parallel data. For example, how will the proposed method reduce the amount of parallel data needed to match the performance of fully parallel data training?

Confidence in this Review

2-Confident (read it all; understood it all reasonably well)